# Evaluation of Social Responsibility of Major Municipal Road Infrastructure—Case Study of Zhengzhou 107 Auxiliary Road Project

**Delei Yang [1], Jiawen Li [1], Jiudong Peng [2], Jun Zhu [3,\*] and Lan Luo [4]**

[1] School of Construction Management and Real Estate, Henan University of Economics and Law, Zhengzhou 450016, China; deleiyang_tju@126.com (D.Y.); jiawen__li@163.com (J.L.)
[2] Henan Urban Planning Institute & Corporation, Zhengzhou 450053, China; ghydqyfy@163.com
[3] Department of Construction Management and Real Estate, Tongji University, Shanghai 200092, China
[4] School of Engineering and Construction, Nanchang University, Nanchang 330031, China; mengling2391@163.com
\* Correspondence: johnsonzhu@tongji.edu.cn

**Abstract:** Social responsibility plays an important role in the sustainable development of major municipal road infrastructure. In this study, a major municipal road infrastructure social responsibility (MMRISR) evaluation indicator system is developed for the comprehensive evaluation of social responsibility. Questionnaires and expert interviews were used to screen the initial indicators of the proposed system. Then, 24 indicators were selected from four dimensions to establish an MMRISR evaluation indicator system. The fuzzy analytic hierarchy process was employed to calculate the weights of each indicator. Finally, the Zhengzhou 107 Auxiliary Road Project was adopted as a case study to test the reliability of the proposed evaluation system. The contribution of this study lies in the provision of a novel indicator system for the social responsibility evaluation of major municipal road infrastructures, thus improving the science of project establishment and decision-making. The proposed social responsibility system can provide an efficient decision-making tool for social responsibility governance, fundamentally promoting the sustainable development of major municipal road infrastructures and the achievement of certain sustainable development goals.

**Keywords:** major municipal road infrastructure; megaproject social responsibility; indicator system; fuzzy analytic hierarchy process; sustainable development

## 1. Introduction

The recent emergence of megaprojects has become a global phenomenon. Megaprojects, such as high-speed railroads, large expressways networks, natural gas pipeline projects, large-span bridges, and large hydropower projects, are major infrastructure projects (MIPs) [1]. MIPs considerably affect national and regional economies, public goods, the environment, politics, and many other aspects of the project lifecycle [2–6]. The continuous progress and expansion of MIPs increasingly requires a consideration of various economic, environmental, and social issues [7–9]. Given the attribute of megaprojects being public goods, their externalities should consider and fulfill social responsibilities [10,11]. Social responsibility plays a key role in improving MIP sustainability and enhancing project performance [12,13]. Megaproject social responsibility (MSR) refers to "the policies and practices of the stakeholders through the whole project lifecycle that reflect responsibilities for the well-being of the wider society" [14]. MSR has become a key factor affecting the sustainable development of MIPs [15]. As a key element of sustainable development, MSR is gradually becoming a new research hotspot in the field of engineering management [1,14]. MSR has also attracted attention from both academics and practitioners [16–18].

Major municipal road infrastructure (MMRI) is defined as the contributive component in MIPs and public service systems [14]. MMRI plays a crucial role in the development of

urbanization and the economy at the present stage [19–21]. In general, MIPs are characterized by basic, nontradable, indivisible, and quasipublic goods, with huge investment and construction scale, wide coverage, complex stakeholders, high dynamic sensitivity, and extensive influence [1,14,22]. In addition to the above characteristics, the most significant feature of MMRI is that it is a "public good" road, which is provided to the public free of charge after project completion. The government and municipal departments are responsible for the maintenance and management of MMRIs. In addition, MMRI directly serves the inner city. Typically located in dense areas that surround residents, MMRI is closely related to the people's daily lives and travels [23,24]. However, numerous problems, including urban dust pollution, noise pollution, and soil and water damage, can easily arise in the presence of MMRIs.

The fulfillment of social responsibility can improve the performance of MMRIs in terms of their environmental, social, economic, and legal aspects [2]. MSR is an increasingly important area in MIPs' sustainable development [7,12,13,25]. Recent developments in the field of sustainability have led to the popularization of major municipal road infrastructure social responsibility (MMRISR), an aspect that is extremely difficult to ignore [14]. The lack of MMRISR negatively impacts a range of other factors, including social development, economic construction, livelihood improvement, and the achievement of sustainable development goals [2,26,27]. For example, the Karakum Canal and the Three Gorges Dam have both caused catastrophic ecosystem degradation. The Three Gorges Dam, the world's largest hydroelectric project and a symbol of China's development of MIPs, has also been fiercely criticized for the threats it poses to the environment, animal species, and migrants [16,28–30]. Therefore, there is an urgent need for MMRISR indicators to improve the fulfillment of MSR. First, existing social responsibility indicator systems usually target corporate social responsibility (CSR) [31], and they cannot fully meet the requirements of MSR [32]. Second, other stakeholders are often not considered and are excluded, except for construction parties. Third, the existing indicators mainly focus on megaprojects [2]. These indicators cannot accurately reflect the specific performance of MSR in a single type of megaprojects. Fourth, existing indicators focus on macrolevel outcomes [25], making it difficult to translate strategic goals into practical MSR actions.

Motivated by the aforementioned discussion, this study proposes an indicator system for evaluating MMRISR. The proposed system includes four dimensions, namely, the political; economic and quality; legal; and environmental and ethical dimensions, with a total of 24 secondary indicators. The fuzzy analytic hierarchy process (FAHP), a quantitative method for dealing with uncertainties and fuzziness in complex problems [23], is employed in this study to build the evaluation indicator system. MMRISR evaluation is an ambiguous and multidimensional complex problem [2,32], and the FAHP approach can be fully applied to this topic. The Zhengzhou 107 Auxiliary Road Project is analyzed as a case study to test the feasibility and practicability of the evaluation indicator system.

The indicator system is designed to respond to the question: "How do we evaluate MMRISR?" This study, which encompasses a holistic view of research, aims to establish a comprehensive scientific evaluation indicator system and provide a decision-making tool for socially responsible governance [33]. This study can fill a gap in MMRISR research, hoping to significantly promote project performance while ensuring the sustainable development of MIPs. An evaluation indicator system of MMRISR based on the FAHP approach is established, thus providing theoretical support for the sustainability evaluation of megaprojects and contributing to the achievement of sustainable development goals.

The rest of this paper is structured as follows. Section 2 describes the research methodology for establishing the MMRISR indicator system. Section 3 presents the results and analysis. Section 4 discusses the indicator system and weighting results. Section 5 summarizes the conclusions and presents the limitations of the study and the potential for future research.

## 2. Literature Review

Sustainability is an extremely important topic in the field of social responsibility from the viewpoint of infrastructure management [13,14,34–36]. Social responsibility in the implementation of MIPs has been investigated from a variety of perspectives. Given the advances in MSR research, there is an urgent need for methods and techniques that enable sustainability assessment and decision-making on important issues of social responsibility. Current research on social responsibility evaluation methods is not suitable for MMRISR due to the different perspectives on social responsibility and the different types of megaprojects. A comprehensive and scientific system of indicators for a single type of megaprojects is urgently needed.

First, indicators originating from the corporate perspective point only to CSR and focus only on aspects of CSR [32,37]. Many scholars, experts, and organizations have re-searched and developed various types of indicator systems, ranging from ethics and morality, to bringing charity to safety issues [38,39]. At the same time, international standards organizations have established international management standards targeting CSR with the aim of improving the motivation of companies to fulfill CSR in order to achieve better social performance, such as ISO 14001, ISO 26000, and SA 8000.

Second, the existing social responsibility evaluation indicator system is only applicable to construction companies [32,40,41]. In this indicator system, most of the stakeholders of megaprojects, such as designers, the public, and decision makers, are excluded and not considered.

Third, from the perspective of project types, these indicators only focus on the overall aspect of megaprojects. There is noticeably poor coverage of the single type of megaprojects. The MSR evaluation conclusions derived from these indicators alone do not accurately reflect the performance status and issues of MSR for a single megaproject type throughout the project life cycle. Since megaprojects involve a variety of fields, different types of megaprojects should use different evaluation indicator systems [2]. Project types are also set in different but specific contexts, and distinct social responsibility evaluation indicator systems for varied types of megaprojects of MMRI have not yet been established.

Fourth, the indicators from the perspective of social responsibility are more focused on the microlevel. However, current social responsibility research tends to focus only on the macrolevel results of national visions and strategic goals [42], with limited research on microlevels [25]. While current megaprojects purposes and national sustainable development strategy frameworks focus more broadly on national aspirations and strategic goals, challenges remain in translating these strategic goals to the microlevel [2]. These findings can hardly be translated into practical actions when each participant at the microlevel needs to be considered [25].

Furthermore, scholars have established indicator systems related to sustainability in different fields based on various methodologies [12,41,43–45]. The industrial ecology approach has been adopted in view of the environmental impacts of products throughout their life cycles [46,47]. Based on the principles of ecology, this approach is applied to achieve industrial ecology through ecological reorganization, ultimately obtaining multiple economic, social, and ecological benefits, and ultimately achieving the sustainable development of human society. In addition, other scholars have used other methods such as group decision-making [48], the LEED green building rating system [49–52], and lifecycle cost analysis [53,54]. In particular, multiple-criteria decision-making (MCDM) is gaining popularity as a comprehensive approach to ensure that all important aspects and interactions are considered [25,55–57]. The unique characteristics of different types of projects, the hierarchical nature of social responsibility, and the multifaceted nature of sustainability objectives should be considered and incorporated into the decision-making process. Based on the combination of fuzzy set theory and hierarchical analysis, the application of FAHP provides significant improvements in the solution of fuzzy and uncertain problems [58–61].

Therefore, this study developed a holistic indictor system based on fuzzy hierarchical analysis to provide a method to effectively evaluate MMRISR.

## 3. Methodology

This study applied various research methods to achieve the research objectives. The research process consisted of five main stages (Figure 1). We first selected the MMPRISR evaluation indicators by conducting a literature analysis of the factors affecting MMPRISR. Second, the indicators were screened on the basis of the initial indicators by means of a questionnaire-based survey and expert interviews. Then, FAHP was used to establish a model for the screened indicators and calculate the weights of each index layer and indicator layer. Finally, the case study was adopted to test the reliability of the indicator system.

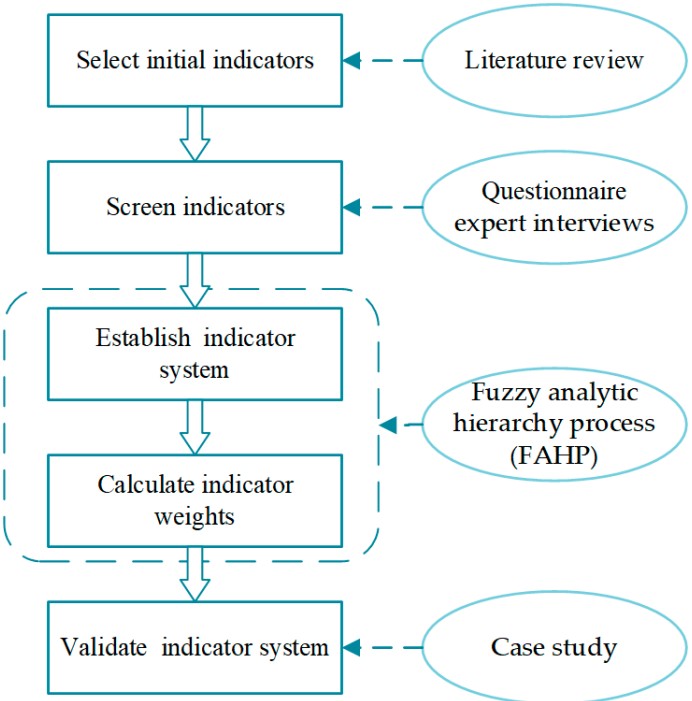

**Figure 1.** Research framework and process.

### 3.1. Selection of the Primary Indicators

#### 3.1.1. Social Responsibility Dimension Determination

The first stage for determining the appropriate and scientific set of MMRISR indicators is to select the initial indicators. Different models of social responsibility, such as the concentric circle model [38], pyramid model [31], and triple bottom-line model [62], have been reported in the literature. On the basis of those studies, social responsibility encompasses the economic, legal, social, environmental, philanthropic, and ethical aspects. One of the most widely used models is Carroll's social responsibility pyramid model [31]. In accordance with the classic pyramid model of social responsibility, this study adopted the social responsibility evaluation indicator system proposed by Lin et al. [2], based on MIPs, and subsequently added social responsibility indicators for depicting the unique background of MMRI. Finally, the social responsibility dimension of MMRI was divided into four areas: economic and quality responsibility, legal responsibility, environmental and ethical responsibility, and political responsibility [2,14,27].

#### 3.1.2. Identification of Indicator Sources

On the basis of the existing literature and the development of social responsibility research theories on MMRI, we reviewed many papers on the project management, social responsibility, and sustainable development of MIPs. The unprecedented worldwide attention to sustainable development and social responsibility has led several projects to aim for certifications showing their ability to conduct sustainable activities following

international standards [2]. Therefore, in addition to the academic literature, this study considered international principles and guidelines as key sources of MSR indicators.

The sources of social responsibility indicators for MMRI are as follows [32]:

(1) Academic literature—A systematic search of relevant databases was conducted by using keywords such as "infrastructure project", "social responsibility", "stakeholder" and "indicator system", and the publication year is between 2001 and 2021;

(2) Relevant international standard systems for the social responsibility guidelines, such as ISO 26000;

(3) Relevant social responsibility reports related to MIPs, such as those issued by enterprises or organizations;

(4) Relevant industry principles and guidelines, such as the Guidelines on Social Responsibility in the Chinese Foreign Contracting Engineering Industry; and

(5) Relevant feasibility study reports, such as prefeasibility study decision reports for MMRI.

### 3.2. Screening for Indicator Optimization

The design of the questionnaires was based on the literature review, case studies, and expert interviews. Due to the limitations of the sample of megaprojects, the questionnaires used in this study were sent to 20 senior managers who had participated in the planning and establishment of MMRI to solicit an expert evaluation of social responsibility indicators at the stage of indicator determination and indicator relative importance judgment. Therefore, the established system of social responsibility evaluation indicators is only for the MMRI field. The specific content of questionnaire is given in Appendices A and B. Meanwhile, social responsibility indicators are highly subjective. Thus, the data sources in this study were strictly controlled during indicator selection. To maximize variability and generalizability, we sought out megaproject experts and senior managers of different sizes, located in different geographic locations, and demonstrating different levels of MSR performance. In particular, we ensured that the interviewed experts had participated in the construction of at least one MMRI. The experts are all highly educated, most of them have a master's degree or a higher educational attainment, and have intermediate or higher titles. The experts also include the following: those with government roles; owners; part of design, construction, supervision, and operation units; and those who had participated in the project establishment, design, construction and operation, and maintenance stages. Since the interviewed experts are senior managers with long experience in the engineering industry, they have more professional experience in this field to ensure the reliability of the survey, and the gender of all the experts is male, which basically conforms to the characteristics of gender imbalance in the engineering field. The background information on the interviewed experts is shown in Table 1. These data proportions basically reflect the actual situation of organizational forms in China. The data indicate that the interviewed experts have experience and knowledge of the issues related to this study, which adds to the credibility of the data. Furthermore, ten experts who had participated in the construction of MMRI were invited to judge the MMRISR evaluation indicator system in terms of the degree of recognition of the indicators.

### 3.3. Establishment of the Indicator System

In accordance with the MMRI perspective, four aspects of the MMRISR indicators were summarized from the existing studies. The MMRISR evaluation indicator system in this study was established by screening social responsibility indicators by means of questionnaires and expert interviews. The indicator system was stratified into three levels as follows. The first level is the target level, which represents the indicator system used to achieve the final purpose of the evaluation. The second level is the component level with four evaluation components: economic and quality responsibility, legal responsibility, environmental and ethical responsibility, and political responsibility. The third level is the factor level, in which each evaluation component can be determined on the basis of several

evaluation factors. Finally, the evaluation indicator system was developed to include the three levels (target, component, and factor levels) and screening indicators [63].

**Table 1.** Background information of interviewed experts.

| Type | Classification | Percentage | Type | Classification | Percentage |
|---|---|---|---|---|---|
| Gender | Male | 100% | Years of work | 1 to 5 years | 10% |
| | Female | 0 | | 6 to 10 years | 70% |
| Age | 21 to 30 years old | 10% | | 11 to 20 years | 10% |
| | 31 to 40 years old | 70% | | More than 20 years | 10% |
| | 41 to 50 years old | 10% | Unit Roles | Government | 10% |
| | >50 years old | 10% | | Owners | 10% |
| Specialties | Engineering Technology | 50% | | Designers | 50% |
| | Engineering Management | 50% | | Constructor | 20% |
| Academic qualifications | Undergraduate | 30% | | Supervisors | 10% |
| | Master | 60% | | Operators | 10% |
| | PhD | 10% | Project Phase | Project stage | 70% |
| Title | Primary | 10% | | Design Phase | 70% |
| | Intermediate | 40% | | Construction Phase | 60% |
| | Advanced | 50% | | Operation and maintenance phase | 10% |

*3.4. Calculation of the Indicator Weights*

FAHP is a multi-indicator evaluation method that combines the analytic hierarchy process (AHP) and the fuzzy comprehensive evaluation (FCE). Fuzzy hierarchical analysis, which is an extension of the basic hierarchical analysis for solving uncertain and ambiguous problems, is a multiple-criteria decision-making (MCDM) method for solving complex problems with uncertainties and ambiguity [64,65]. MMRI is a complex system. The evaluation indexes of MMRISR are similar to hierarchical relationships in that they involve more factors and have clear hierarchical relationships [2]. The majority of the MMRISR evaluation indicators are qualitative indicators, and the measurement language is fuzzy [66]. However, unlike economic evaluation indicators, the MMRISR indicators are difficult to quantify into specific numbers and formulas, and their language belongs to the domain of fuzzy boundaries [67,68]. Therefore, a numerical analysis was applied in the current research to calculate the weight priority of the MMRISR indicators, and FAHP was used for multi-criteria decision-making [69,70]. The steps involved in the weight calculation include the following:

Step 1: Establish the fuzzy judgment matrices.

In fuzzy hierarchical analysis, the fuzzy judgment matrix can be derived by instructing experts to compare indicators according two parametric importance. The two-by-two comparison judgment between the factors is expressed quantitatively using the importance of one factor over another factor. The fuzzy judgment matrix is given by $A = (A_{ij})_{n*n}$, where $A$ is the fuzzy complementary judgment matrix, and $a_{ij}$ is the degree of affiliation (i.e., "more important than"). The magnitude of this degree is denoted by a fuzzy judgment scale with values ranging from 0.1 to 0.9. From the importance linguistic scale, the factors can then be compared with each other, and the following fuzzy complementary judgment matrix is obtained:

$$A = \begin{bmatrix} a_{11} & a_{12} & \cdots & a_{1n} \\ a_{21} & a_{22} & \cdots & a_{2n} \\ \cdots & \cdots & \cdots & \cdots \\ a_{n1} & a_{n2} & \cdots & a_{nn} \end{bmatrix}$$

Step 2: Calculate the relative importance of the factors.

In general, FAHP requires multiple experts to perform the evaluation, which can be denoted as:

$$A_i^{(l)} = \sum_{k=1}^{n} a_{ik}^{(l)}, i = 1, 2, \ldots n, l = 1, 2, \ldots, s \tag{1}$$

In real decision-making, the obtained fuzzy judgment matrix is contradictory, due to the complexity of parameters and the one-sidedness of understanding these parameters. Hence, knowing the consistency of a fuzzy judgment matrix is essential. According to the definition of the fuzzy consistency matrix, the mathematical transformation is performed as follows:

$$b_{ij}^{(l)} = \frac{a_i^{(l)} - a_j^{(l)}}{2(n-1)} + 0.5, l = 1, 2, \dots, s. \tag{2}$$

The fuzzy consistency matrix is obtained as follows:

$$B = \left( b_{ij}^{(l)} \right)_{n \times n}, (l = 1, 2, \dots, s). \tag{3}$$

Suppose the number of experts is *m*, and each expert is given the same weight of $\lambda_1 = \lambda_2 = \dots \lambda_\infty = \delta$. Then,

$$\omega_i = \frac{\sum_{i=1}^n \sum_{j=1}^n \lambda b_{ij}^{(l)} + \frac{n}{2} - 1}{n(n-1)} \tag{4}$$

The relative weights of each indicator in the matrix can be obtained by $\omega_i$.

Step 3: Rank the overall importance of each factor.

The relative importance of the low-level factors with respect to the high-level factors are calculated. Then, the importance is ranked on the basis of the above steps to determine the position of each factor in the overall evaluation index system.

### 3.5. Comprehensive Evaluation

Step 1: Establish the single-factor fuzzy judgment matrix.

Establish a single-factor evaluation rubric set $V = \{V1, V2, \dots, Vm\}$, for the qualitative indicators, and let the experts involved in the evaluation rate the indicators $U_i$. Count the frequency of evaluation results belonging to rank $V_i M_{ij}$. Calculate:

$$r_{v_j}(D_i) = \frac{M_{ij}}{N} \tag{5}$$

where $M_{ij}$ is the number of evaluation results belonging to $V_i$ (number of times), and $N$ is the number of experts involved in the evaluation. $r_{v_j}(D_i)$ is $D \in V_i$ (affiliation degree).

Then, the indicator $U_i$ (affiliation function) is calculated as:

$$r_{v_j} = \frac{r_{v_1}(D_i)}{V_1} + \frac{r_{v_2}(D_i)}{V_2} + \cdots + \frac{r_{v_m}(D_i)}{V_m}. \tag{6}$$

By collecting, organizing, and counting the questionnaires, the one-factor judgment matrix (*R* of *U* to *V*) can be established as:

$$R = \begin{bmatrix} r_{11} & r_{12} & \cdots & r_{1m} \\ r_{21} & r_{22} & \cdots & r_{2m} \\ \vdots & \vdots & \ddots & \vdots \\ r_{n1} & r_{n2} & \cdots & r_{nm} \end{bmatrix}$$

Step 2: Establish the comprehensive evaluation model.

Fuzzy-transform the single-factor evaluation matrix with the set of weights, and set its weight as $W = (w_1, w_2, \ldots, w_n)$. Obtain the FCE model by calculating:

$$r = W \cdot R = (w_1, w_2, \ldots, w_n) \begin{bmatrix} r_{11} & r_{12} & \cdots & r_{1m} \\ r_{21} & r_{22} & \cdots & r_{2m} \\ \vdots & \vdots & \ddots & \vdots \\ r_{n1} & r_{n2} & \cdots & r_{nm} \end{bmatrix} = (r_1, r_2, \ldots, r_m) \tag{7}$$

The fuzzy synthetic operation in Formula (7) is the ordinary matrix multiplication method (weighted average method). This model allows each factor to contribute to the comprehensive evaluation, objectively reflecting the entire status of the evaluation object.

## 4. Analysis and Results

### 4.1. Analysis of Indicator Screening

From the questionnaire survey, 20 experts were allowed to score 32 social responsibility indicators. The normalized and ranked results are shown in Table 2. The indicators whose normalized values were equal to or higher than 0.4 were selected as the key measures of the MMRISR evaluation.

The rationality of the indicator system was also confirmed with the experts, and the reasons for excluding the indicators with normalized values lower than 0.4 were analyzed. The responses included the following explanations. First, the indicator already has a clear responsible subject, and the relevant parties are already relatively good in certain aspects (e.g., compensation for land acquisition and demolition, reasonable use of construction resources, active organization of public participation, technological innovation and application, and road maintenance). Second, other indicators (project reporting compliance, project reporting independence and impartiality, and public opinion supervision) are less important than the other indicators. Therefore, 24 indicators for social responsibility evaluation were screened for the establishment of the FAHP model.

**Table 2.** Ranking of MMRISR indicators.

| Number | Indicators | Average Value | Normalization | Ranking |
|---|---|---|---|---|
| 1 | Meets the needs of residents' life and travel | 4.9 | 1.00 | 1 |
| 2 | Public event emergency response | 4.9 | 1.00 | 1 |
| 3 | Quality and safe construction of road projects | 4.9 | 1.00 | 1 |
| 4 | Water, noise, dust, and other pollution control | 4.8 | 0.94 | 4 |
| 5 | Regional transportation network construction | 4.7 | 0.88 | 5 |
| 6 | Proper handling of the relationship between people, vehicles, roads, and the environment | 4.7 | 0.88 | 5 |
| 7 | Project technical feasibility decision | 4.7 | 0.88 | 5 |
| 8 | Raising awareness of environmental protection | 4.7 | 0.88 | 5 |
| 9 | Road construction cost and schedule control | 4.6 | 0.82 | 9 |
| 10 | Design, construction, and operation comply with transportation industry specifications and legal requirements | 4.6 | 0.82 | 9 |
| 11 | Environmental ecological and cultural protection along the road | 4.6 | 0.82 | 9 |
| 12 | Consideration of the impact of the proposed road on existing tracks | 4.5 | 0.76 | 12 |
| 13 | Protecting the rights and interests of participating employees | 4.5 | 0.76 | 12 |
| 14 | Coordinate the sequence of road construction and pipeline construction | 4.5 | 0.76 | 12 |
| 15 | Sound engineering project governance mechanism | 4.4 | 0.71 | 15 |
| 16 | Interoperability with the surrounding environment | 4.4 | 0.71 | 15 |
| 17 | Monitoring and reporting wrongdoing | 4.4 | 0.71 | 15 |
| 18 | Maintaining relationships with the surrounding community | 4.3 | 0.65 | 18 |
| 19 | Project economic feasibility decision | 4.3 | 0.65 | 18 |
| 20 | Project operating costs and safety assurance | 4.3 | 0.65 | 18 |
| 21 | Effective coordination between the main government departments | 4.2 | 0.59 | 21 |
| 22 | Information disclosure | 4.2 | 0.59 | 21 |

| Number | Indicators | Average Value | Normalization | Ranking |
|---|---|---|---|---|
| 23 | Engineering anticorruption | 4 | 0.47 | 23 |
| 24 | Effective regulation | 4 | 0.47 | 23 |
| 25 | Rational use of construction resources | 3.8 | 0.35 | 25 |
| 26 | Road maintenance | 3.8 | 0.35 | 25 |
| 27 | Project coverage compliance | 3.8 | 0.35 | 25 |
| 28 | Technology innovation and application | 3.8 | 0.35 | 25 |
| 29 | Project reporting independence and impartiality | 3.8 | 0.35 | 25 |
| 30 | Compensation for land acquisition and relocation | 3.5 | 0.18 | 30 |
| 31 | Public opinion monitoring | 3.3 | 0.06 | 31 |
| 32 | Actively organizes public participation | 3.2 | 0.00 | 32 |

### 4.2. Establishment of the Indicator System

The MMRISR evaluation indicator system was divided into three levels: target, component, and factor levels. Social responsibility was divided into four dimensions: economic and quality, legal, environmental and ethical, and political. After the screening, 24 indicators were retained to establish the MMRISR evaluation indicator system. Figure 2 shows the hierarchy model of the MMRISR evaluation indicators.

### 4.3. Analysis of Indicator Weighting

The questionnaire data were used for the weighting. The fuzzy judgment matrices of the first-level social indicators $A$ and second-level social responsibility indicators $U_1$, $U_2$, $U_3$, and $U_4$ were obtained using Formula (1).

$$A = \begin{bmatrix} 0.5 & 0.59 & 0.58 & 0.57 \\ 0.41 & 0.5 & 0.57 & 0.58 \\ 0.42 & 0.43 & 0.5 & 0.6 \\ 0.43 & 0.42 & 0.4 & 0.5 \end{bmatrix}$$

$$U_1 = \begin{bmatrix} 0.5 & 0.43 & 0.63 & 0.68 & 0.65 & 0.63 & 0.54 \\ 0.57 & 0.5 & 0.62 & 0.63 & 0.65 & 0.57 & 0.55 \\ 0.37 & 0.38 & 0.5 & 0.58 & 0.55 & 0.54 & 0.52 \\ 0.32 & 0.37 & 0.42 & 0.5 & 0.55 & 0.53 & 0.51 \\ 0.35 & 0.35 & 0.45 & 0.45 & 0.5 & 0.58 & 0.51 \\ 0.37 & 0.43 & 0.46 & 0.47 & 0.42 & 0.5 & 0.58 \\ 0.46 & 0.45 & 048 & 0.49 & 0.49 & 0.42 & 0.5 \end{bmatrix}$$

$$U_2 = \begin{bmatrix} 0.5 & 0.45 & 0.63 & 0.47 & 0.53 & 0.53 \\ 0.55 & 0.5 & 0.58 & 0.51 & 0.59 & 0.53 \\ 0.37 & 0.42 & 0.5 & 0.51 & 0.56 & 0.5 \\ 0.53 & 0.49 & 0.49 & 0.5 & 0.59 & 0.58 \\ 0.47 & 0.41 & 0.44 & 0.41 & 0.5 & 0.54 \\ 0.47 & 0.47 & 0.5 & 0.42 & 0.46 & 0.5 \end{bmatrix}$$

$$U_3 = \begin{bmatrix} 0.5 & 0.61 & 0.53 \\ 0.39 & 0.5 & 0.56 \\ 0.47 & 0.44 & 0.5 \end{bmatrix}$$

$$U_4 = \begin{bmatrix} 0.5 & 0.57 & 0.56 & 0.54 & 0.48 & 0.53 & 0.47 & 0.56 \\ 0.43 & 0.5 & 0.53 & 0.54 & 0.5 & 0.56 & 0.51 & 0.55 \\ 0.44 & 0.47 & 0.5 & 0.56 & 0.47 & 0.53 & 0.47 & 0.57 \\ 0.46 & 0.46 & 0.44 & 0.5 & 0.5 & 0.59 & 0.5 & 0.53 \\ 0.52 & 0.5 & 0.53 & 0.5 & 0.5 & 0.56 & 0.46 & 0.54 \\ 0.47 & 0.44 & 0.47 & 0.41 & 0.44 & 0.5 & 0.45 & 0.57 \\ 0.53 & 0.49 & 0.53 & 0.5 & 0.54 & 0.55 & 0.5 & 0.55 \\ 0.44 & 0.45 & 0.43 & 0.47 & 0.46 & 0.43 & 0.45 & 0.5 \end{bmatrix}$$

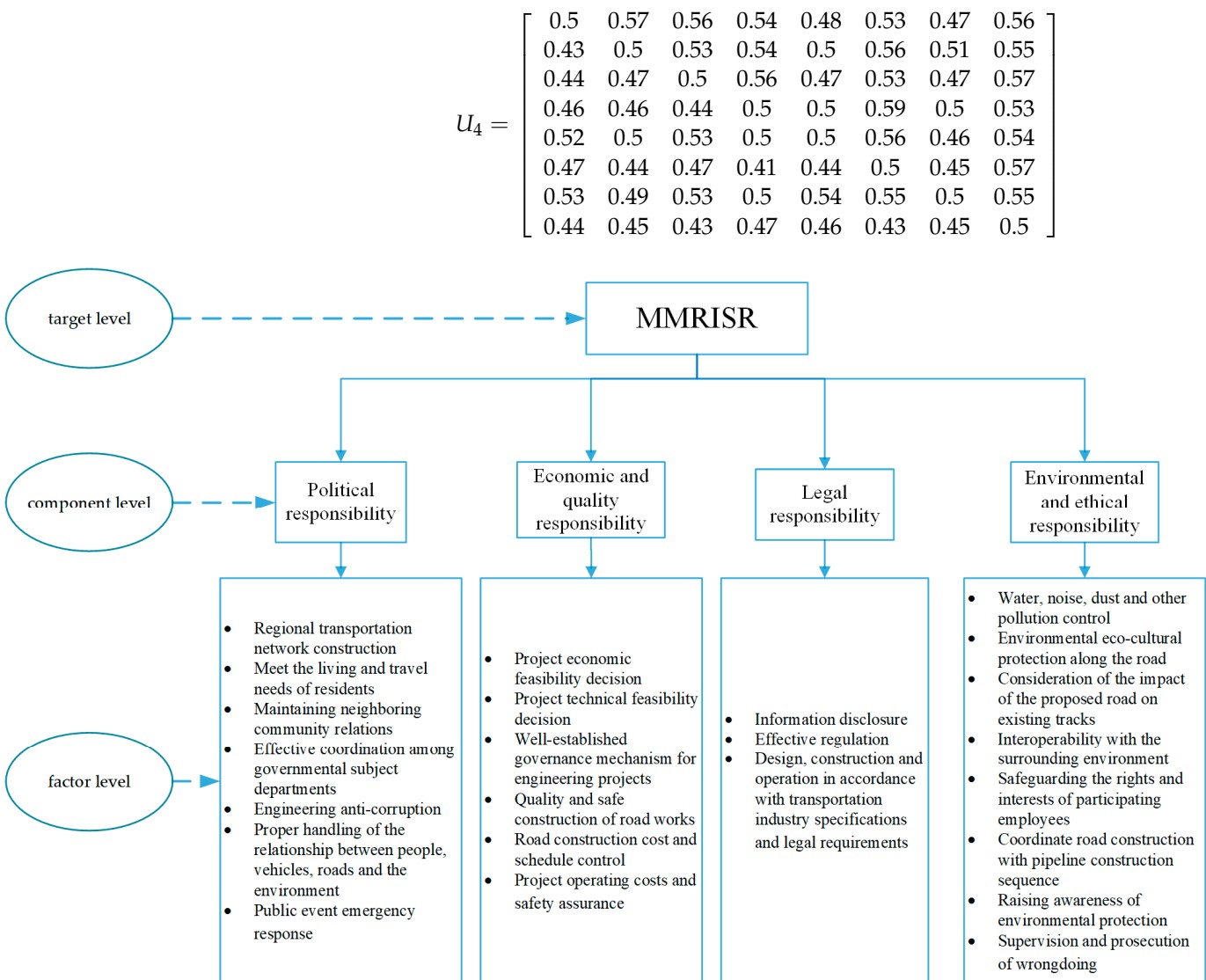

**Figure 2.** Hierarchy model of the MMRISR evaluation indicators.

The fuzzy mathematical transformation was conducted according to Formula (2), and the fuzzy consistent judgment matrix of $M_1$, $N_1$, $N_2$, $N_3$, and $N_4$ was obtained using Formula (3). Consider that the fuzzy judgment matrix A is composed of level-1 social responsibility indicators. The weight calculation process is as follows.

The mathematical transformation is performed using the formula of fuzzy judgment matrix $A$ to obtain the fuzzy consistency matrix $M_1$.

$$M_1 = \begin{bmatrix} 0.5 & 0.53 & 0.55 & 0.58 \\ 0.47 & 0.5 & 0.52 & 0.55 \\ 0.45 & 0.48 & 0.5 & 0.53 \\ 0.42 & 0.45 & 0.47 & 0.5 \end{bmatrix}$$

The weights subsequently are calculated using Formula (4), i.e., $\omega_1 = (0.2633, 0.2533, 0.2472, 0.2361)^T$.

For the weight coefficients of each evaluation indicator at the criterion level, the following results were obtained: political responsibility (0.2633), economic and quality responsibility (0.2533), legal responsibility (0.2472), and environmental and ethical responsibility (0.2361).

Similarly, the fuzzy judgment matrix within the criterion and the relative importance of each index was derived from the questionnaire data. The final results are shown in Table 3.

The findings indicate that the MMRI should comprehensively consider social responsibility with respect to the political, economic and quality, legal, and environmental and ethical aspects.

For political responsibility, meeting the living and travel needs of residents and building a regional transportation network obtained the top two ranks. This finding reflects the purpose of the MMRI, which is to provide services to the public as a means of improving the quality of life of residents [71]. In this regard, the government plays a key role in social responsibility management because it is the subject of political responsibility. Furthermore, the government has the final decision-making power on the construction of MMRI [72]. If the government's exercise of public power ignores regulation and restraint, then it will cause significant irreparable damage to social stability, national property, and people's life safety. The implementation of anticorruption measures is also important in avoiding potential quality problems in engineering and safety accidents [73]. In addition, the government's emergency response to public events also serves as a guarantee for maintaining social stability.

For economic and quality responsibility, the indicators of project technical feasibility and road project quality and safety construction ranked the highest. Technical decision-making, quality control, and safe construction represent the basic objectives of MMRI [9]. Technical errors in preproject decision-making or quality defects during construction will lead to construction accidents or human casualties. In addition, as MMRI is a public good, it is a nonprofit project invested in by the government to improve the people's accessibility and accelerate urban development [10]. However, given the financial restrictions of the government, scientific and reasonable economic decisions can be implemented in the long run to support the construction of non-operating MIPs, achieve the sustainable development of MIPs, and fulfill the sustainability requirements of the MMRISR.

For legal responsibility, it includes the code of conduct of relevant participants with respect to the design, construction, and operation norms in the transportation industry and the related legal requirements [14]. Breaches in legal defense indicate many types of opportunistic behavior [74]. Promoting information disclosure and effective supervision of MMRI is a good approach to realizing the governance of MMRISR.

For environmental and ethical responsibility, strengthening the pollution control of water, noise, and dust and raising awareness of environmental protection are the most popular topics at present [18,75]; in China, they correspond to the increasing emphasis on the concept of sustainable development [76,77]. Only by respecting nature and protecting the environment can the people and society achieve sustainable development [78].

**Table 3.** Weight results for MMRISR indicator system.

| Target Level | Component Level | Factor Level | Combined Weights |
|---|---|---|---|
| Social responsibility evaluation of major municipal road infrastructure | Political responsibility (0.2633) | Regional transportation network construction (0.1506) | 0.03966 |
| | | Meets the living and travel needs of residents (0.1522) | 0.03977 |
| | | Maintaining neighboring community relations (0.1420) | 0.03739 |
| | | Effective coordination among governmental subject departments (0.1387) | 0.03652 |
| | | Engineering anticorruption (0.1386) | 0.03648 |
| | | Proper handling of the relationship between people, vehicles, roads, and the environment (0.1391) | 0.03663 |
| | | Public event emergency response (0.1399) | 0.03685 |
| | Economic and quality responsibility (0.2533) | Project economic feasibility decision (0.1689) | 0.04277 |
| | | Project technical feasibility decision (0.1719) | 0.04353 |
| | | Well-established governance mechanism for engineering projects (0.1639) | 0.04151 |
| | | Quality and safe construction of road works (0.1701) | 0.04308 |
| | | Road construction cost and schedule control (0.1621) | 0.04105 |
| | | Project operating costs and safety assurance (0.1631) | 0.04130 |
| | Legal responsibility (0.2472) | Information disclosure (0.3508) | 0.08673 |
| | | Effective regulation (0.3271) | 0.08086 |
| | | Design, construction, and operation in accordance with transportation industry specifications and legal requirements (0.3221) | 0.07962 |
| | Environmental and ethical responsibility (0.2361) | Water, noise, dust and other pollution control (0.1271) | 0.03002 |
| | | Environmental eco-cultural protection along the road (0.1262) | 0.02980 |
| | | Consideration of the impact of the proposed road on existing tracks (0.1251) | 0.02954 |
| | | Interoperability with the surrounding environment (0.1248) | 0.02946 |
| | | Safeguarding the rights and interests of participating employees (0.1261) | 0.02978 |
| | | Coordinate road construction with pipeline construction sequence (0.1224) | 0.02891 |
| | | Raising awareness of environmental protection (0.1269) | 0.02997 |
| | | Supervision and prosecution of wrongdoing (0.1212) | 0.02862 |

## 5. Case Study

### 5.1. Case Background

The Zhengzhou 107 Auxiliary Road Project is located in Zhengzhou City, Henan Province, China. The auxiliary road is an eight-lane road in both north and south directions. Its total length is approximately 20 km and comprises six interchanges. The auxiliary road project is one of the megaprojects classified as a MIP. The Zhengzhou 107 Auxiliary Road Project is a fast-running lane and has been put into operation. A social responsibility evaluation of this project will allow for subsequent megaprojects to be sustainable.

### 5.2. Acquisition and Quantification of the Evaluation Indicator Data

All evaluation results should be accurate and reliable. In this study, questionnaires were issued to design and construction experts who had participated in constructing the Zhengzhou 107 Auxiliary Road Project. Ten valid questionnaires were received. The questionnaire entailed a five-level rubric set in which $V = \{excellent, good, fair, poor, very\ poor\}$ $= \{V1, V2, V3, V4, V5\}$. The single-factor evaluation was conducted in the questionnaire format to derive the rubric for each factor. The results of the social responsibility evaluation of the Zhengzhou 107 Auxiliary Road Project were obtained by collecting, organizing, and counting the questionnaires.

### 5.3. Application of the Evaluation Model

The affiliation degree of each index is calculated using Formula (6). The results are shown in Table 4.

**Table 4.** MMRISR evaluation indicator survey statistics.

| Indicators | | Evaluation Status | | | | |
|---|---|---|---|---|---|---|
| | | Excellent | Good | General | Poor | Very Poor |
| Political responsibility | Regional transportation network construction | 0.8 | 0.1 | 0.1 | 0 | 0 |
| | Meets the needs of residents' life and travel | 0.9 | 0.1 | 0 | 0 | 0 |
| | Maintaining relationships with the surrounding community | 0.2 | 0.7 | 0.1 | 0 | 0 |
| | Effective coordination between the main government departments | 0.3 | 0.4 | 0.3 | 0 | 0 |
| | Engineering anticorruption | 0.2 | 0.4 | 0.4 | 0 | 0 |
| | Proper handling of the relationship between people, vehicles, roads, and the environment | 0.2 | 0.2 | 0.2 | 0.3 | 0.1 |
| | Public event emergency response | 0.1 | 0.3 | 0.6 | 0 | 0 |
| Economic and Quality Responsibility | Project economic feasibility decision | 0.1 | 0.8 | 0.1 | 0 | 0 |
| | Project technical feasibility decision | 0.3 | 0.2 | 0.3 | 0.2 | 0 |
| | Sound engineering project governance mechanism | 0.1 | 0.4 | 0.4 | 0.1 | 0 |
| | Quality and safe construction of road projects | 0.5 | 0.1 | 0.4 | 0 | 0 |
| | Road construction cost and schedule control | 0.1 | 0.4 | 0.4 | 0.1 | 0 |
| | Project operating costs and safety assurance | 0.1 | 0.5 | 0.4 | 0 | 0 |
| | Information disclosure | 0.2 | 0.1 | 0.7 | 0 | 0 |
| Legal Liability | Effective regulation | 0.2 | 0.8 | 0 | 0 | 0 |
| | Design, construction, and operation comply with transportation industry specifications and legal requirements | 0.2 | 0.6 | 0.2 | 0 | 0 |
| Environmental and Ethical Responsibility | Water, noise, dust, and other pollution control | 0.3 | 0.3 | 0.4 | 0 | 0 |
| | Environmental ecological and cultural protection along the road | 0.2 | 0.7 | 0.1 | 0 | 0 |
| | Consideration of the impact of the proposed road on existing tracks | 0.2 | 0.7 | 0.1 | 0 | 0 |
| | Interoperability with the surrounding environment | 0.3 | 0.1 | 0.3 | 0.2 | 0.1 |
| | Protecting the rights and interests of participating employees | 0.2 | 0.3 | 0.3 | 0.2 | 0 |
| | Coordinate the sequence of road construction and pipeline construction | 0.2 | 0.2 | 0.4 | 0 | 0.2 |
| | Raising awareness of environmental protection | 0.1 | 0.4 | 0.3 | 0.2 | 0 |
| | Monitoring and reporting wrongdoing | 0.2 | 0.3 | 0.4 | 0.1 | 0 |

The single-factor fuzzy judgment matrix ($R$ from $U$ to $V$) can be established using the index affiliation table. The comprehensive evaluation results of the model are calculated using Formula (7), where $W$ is the weighted weight of the social responsibility evaluation indexes obtained via FAHP.

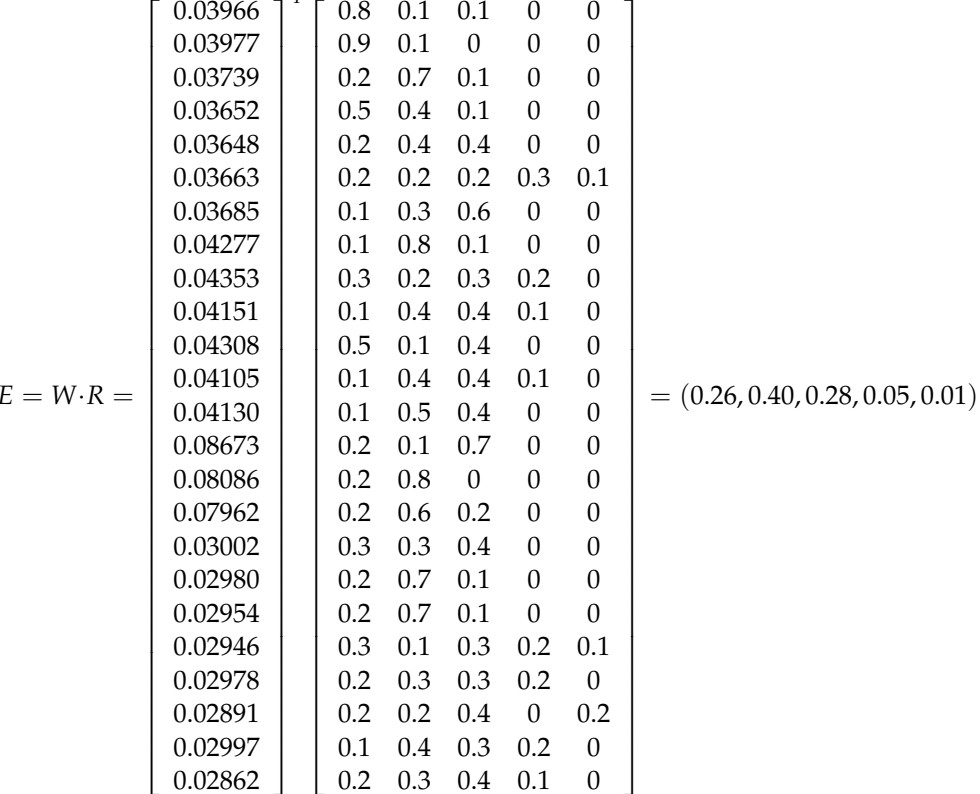

$$E = W \cdot R = \begin{bmatrix} 0.03966 \\ 0.03977 \\ 0.03739 \\ 0.03652 \\ 0.03648 \\ 0.03663 \\ 0.03685 \\ 0.04277 \\ 0.04353 \\ 0.04151 \\ 0.04308 \\ 0.04105 \\ 0.04130 \\ 0.08673 \\ 0.08086 \\ 0.07962 \\ 0.03002 \\ 0.02980 \\ 0.02954 \\ 0.02946 \\ 0.02978 \\ 0.02891 \\ 0.02997 \\ 0.02862 \end{bmatrix}^T \begin{bmatrix} 0.8 & 0.1 & 0.1 & 0 & 0 \\ 0.9 & 0.1 & 0 & 0 & 0 \\ 0.2 & 0.7 & 0.1 & 0 & 0 \\ 0.5 & 0.4 & 0.1 & 0 & 0 \\ 0.2 & 0.4 & 0.4 & 0 & 0 \\ 0.2 & 0.2 & 0.2 & 0.3 & 0.1 \\ 0.1 & 0.3 & 0.6 & 0 & 0 \\ 0.1 & 0.8 & 0.1 & 0 & 0 \\ 0.3 & 0.2 & 0.3 & 0.2 & 0 \\ 0.1 & 0.4 & 0.4 & 0.1 & 0 \\ 0.5 & 0.1 & 0.4 & 0 & 0 \\ 0.1 & 0.4 & 0.4 & 0.1 & 0 \\ 0.1 & 0.5 & 0.4 & 0 & 0 \\ 0.2 & 0.1 & 0.7 & 0 & 0 \\ 0.2 & 0.8 & 0 & 0 & 0 \\ 0.2 & 0.6 & 0.2 & 0 & 0 \\ 0.3 & 0.3 & 0.4 & 0 & 0 \\ 0.2 & 0.7 & 0.1 & 0 & 0 \\ 0.2 & 0.7 & 0.1 & 0 & 0 \\ 0.3 & 0.1 & 0.3 & 0.2 & 0.1 \\ 0.2 & 0.3 & 0.3 & 0.2 & 0 \\ 0.2 & 0.2 & 0.4 & 0 & 0.2 \\ 0.1 & 0.4 & 0.3 & 0.2 & 0 \\ 0.2 & 0.3 & 0.4 & 0.1 & 0 \end{bmatrix} = (0.26, 0.40, 0.28, 0.05, 0.01)$$

*5.4. Analysis of the Evaluation Results*

The obtained FCE result of the affiliation vector E suggests a certain gap in the affiliation values of each result level; the maximum affiliation principle can be used to discriminate the values. According to the principle of maximum affiliation discrimination, the result level with a large affiliation is determined as the final evaluation level. The maximum value is 0.40; therefore, the MMRI final evaluation result is 0.40. The results of the MMRISR evaluation are divided into four grades. As shown in Table 5, the MMRISR evaluation result of the Zhengzhou 107 Auxiliary Road Project is "good," implying that the governance of the MSR can be improved.

**Table 5.** The rank of evaluation results.

| Evaluation results | 0–0.1 | 0.1–0.2 | 0.2–0.3 | 0.3–0.4 | 0.4–0.5 |
|---|---|---|---|---|---|
| Rank | Very poor | Poor | General | Good | Excellent |

In terms of political responsibility, although the planning and establishment of the project can fully build a regional transportation network and meet the needs of public life and travel, shortcomings are apparent. First, the project managers did not conduct a comprehensive planning before implementation, which led to the increase in various costs during the construction period. Second, the road planning was not perfect, and the relation of people, vehicles, roads, and the environment was not properly handled. A secondary design was consequently implemented during the construction of the project, which led to the delay of the construction period. Third, the organization and coordination of government departments is poor. For example, for the railroad-crossing sections and military

facilities along the route, various departments overlooked their responsibilities, resulting in a huge waste of resources. Problems related to economic and quality responsibility were also observed during project implementation. Oftentimes, quality issues are concealed by economic benefits, which is rooted in the orientation of maximizing all benefits [74]. For example, the stakeholders only consider the direct cost of the project for the cost-saving measures, whereas the indirect cost and social cost are ignored. Furthermore, the contractor used "jerry-built materials" for profit, leading to serious quality problems. Regarding the legal responsibility aspect, all parties involved in the construction can restrain one's own behavior according to the legal specifications and design requirements. However, the accountability mechanism of the project is inappropriate. As delineations of responsibilities are lacking, the penalties pertaining to the liability problems are often weak to create a warning effect [27]. The environmental aspects of the project were controlled by dust control during implementation, which greatly improved the environmental pollution problem as opposed to the schemes of previous projects [14]. On the one hand, the measures are mandatory and ensured by the environmental protection department of the government. On the other hand, the project participants seldom took the initiative to fulfill their environmental responsibilities [15], and their awareness remained at a low level.

The aforementioned problems reveal certain shortcomings in MMRISR. By further reforming and improving the decision-making mechanism, the negative effects related to the lack of social responsibility can be greatly improved—for example, by consulting with experts and people from all walks of life, increasing the transparency of decision-making, and raising the awareness of social responsibility among citizens. Undeniably, the project construction has improved the mobility and life of the residents, helped to build a good relationship with the community, and increased the resilience of the MIPs. The positive benefits of the project will more than compensate for the damage to the surrounding environment and livelihoods, ultimately achieving the goal of sustainable development.

The MMRISR evaluation result of Zhengzhou 107 Auxiliary Road Project is generally consistent with the project situation learned through the expert interviews, thus verifying the reliability and usefulness of the proposed MMRISR indicator system.

## 6. Discussion and Implication

### 6.1. Discussion

To date, an adequate and comprehensive MMRISR evaluation indicator system does not exist. The system could have addressed MMRISR performance in a comprehensive and detailed manner and translated the strategic objectives into practical actions. Our research has proposed an MMRISR indicator system that builds upon the existing indicator framework for a megaproject's social responsibility by developing a conceptual framework embedded in MMRI. This study focuses on the social responsibility evaluation of major municipal road infrastructure, and the theoretical construction and empirical results verify the reliability and practicality of the system.

In terms of the four dimensions, the experts believe that the core of the evaluation of MMRISR is "political responsibility". MMRI not only assumes the role of regional network building in the political aspect but can also maintain the surrounding community's relations and fight corruption in engineering projects. The main body of political responsibility relies on the government. Although the four dimensions have similar weight values, the weight of political responsibility is higher than those of the other elements. The analytic results showed that political responsibility represents the highest priority in China's unique government–market institutional context [15]. Given the political connections, major engineering infrastructure projects are usually recognized as a measure of political performance [74], further indicating that political responsibility requires more emphasis. Through the successful completion of MIPs, the participants refer to their desire for political promotions, political access, political appointments, higher levels of government support, and connections to the government [79,80]. As a result, it is possible to improve information sharing and communication between MIPs participants and the government,

gain access to potential market opportunities, and help them to benefit from incentives and subsidies in the future [81].

In terms of the 24 indicators, among the top factors, "information disclosure" is necessary for evaluating MMRISR. Meanwhile, "effective regulation" and "design, construction, and operation in accordance with transportation industry specifications and legal requirements," belonging to the legal responsibility classification, are also imperative for evaluating MMRISR. This result suggests that legal responsibility can be fulfilled through externalities, such as public and media supervision [79,82,83]. The participants could restrain their behavior by focusing on the legal norms and design requirements, but the accountability mechanism is generally faulty [14]. The phenomenon of unclear division of responsibility has caused parallel management and multiple management, leading to the emergence of liability problems caused by the loss of either an equal share of responsibility or the law. Subsequently, unlawful practice is difficult to punish, and the problem at the onset could not be solved. Meanwhile, the public is increasingly concerned about its impact on health and well-being [84]. Therefore, design, construction, and operation should not only comply with industry norms and legal requirements but also achieve effective public and media supervision, information disclosure, and public opinion monitoring, as these indicators are essential for the fulfillment of MMRISR [85]. For example, deficiencies in social responsibility can be reduced by regularly publishing social responsibility reports. In addition, improve the relevant laws. These measures are very important to promote the MMRISR fulfillment. The public and media can help to effectively monitor certain concerns [86,87]. Most of the major engineering social responsibility infrastructure projects remain weak in terms of awareness of legal responsibility. The research results show that information disclosure, effective supervision, and compliance with legal requirements to improve regulation of MMRISR should be the focus of MMRISR fulfillment. Furthermore, MMRI should be built under strict supervision and information disclosure, adhering to the concept of social responsibility and sustainability.

Economic and quality responsibility is also necessary in MMRISR evaluation. The weights of all indicators pertaining to economic and quality responsibility are also relatively high. This scenario indicates that the iron triangle (cost, time, and scope) of traditional engineering project management remains to be a key indicator of preproject decision-making and engineering performance measurement [9,88–90], as cost overruns on MMRI projects are common [88,89,91,92]. Furthermore, the technical feasibility decision at the project stage and the quality and safety construction of road projects are a common concern. If the lack of social responsibility in the aspects of economy and quality leads to quality-related accidents, then waste of resources and loss of social welfare will likely occur [14]. In terms of environment and ethics, pollution control and environmental protection have been taken seriously by all parties participating in the construction [18,77], but the awareness of ethical responsibilities, such as employee rights protection and monitoring and reporting of negligence, fraud, bribery, and dishonest wrongdoings, is relatively weak [78,93,94].

MSR research is being conducted worldwide and has been gaining more and more attention from scholars as it plays an important role in sustainable development. However, due to the different conditions of each country, each country needs to establish an MMRISR evaluation index system according to its own specific needs. Based on the unique government–market dichotomy political context in China, megaprojects are not only public infrastructure, but also a political task. Therefore, the MMRISR evaluation indictor system must include meeting residents' living and travel needs, maintaining surrounding community relations, and properly handling the relationships between people, vehicles, roads, and the environment. In addition, the MMRISR development evaluation indictor system places more emphasis on simple implementation and effective evaluation, and is easy to promote in decision-making and construction units. For example, assessing the quality of road works and road construction cost system are practical measures.

The experts and senior managers have fully considered the evaluation indicators, thus ensuring the reliability and applicability of the evaluation results. The MMRISR

indicator system provides a useful reference for future MMRI construction. Instead of just pursuing fast construction, high yield and low cost, we should focus on regional economic development, ecological protection, community stability, and sustainable development. According to the actual situation in China, policy makers should fully consider the enforceability of policies as well as substantial incentives, such as favorable tax policies, loan support, and more project support from the government, in order to stimulate the willingness to actively fulfill MMRISR. This study provides an in-depth analysis of the experts' assessments. By summarizing the existing literature, preliminary evaluation indicators were screened: then, based on the characteristics and needs of MMRI projects, an MMRISR evaluation indicator system suitable for China was identified through FAHP, which provides a useful reference for promoting sustainable development strategies for major municipal infrastructure in China.

### 6.2. Management Implication

This study entails management implications for project decisionmakers in MIP management. First, the feasibility study report, which is often used as the basis for the construction project, is a key document in the early stages. However, social responsibility as a component of the feasibility study often involves only a short qualitative description of social responsibility; it neither draws attention to stakeholders nor achieves the real evaluation purpose of social responsibility [95,96]. In this study, we recommend improvements to the feasibility study evaluation mechanism by including a social responsibility evaluation in the prefeasibility study report to fundamentally strengthen the social responsibility discussion.

Second, considering the positive role of MMRISR, stakeholders of MIPs are encouraged to strive for MMRISR in practice to achieve the sustainable development of MIPs through economic and quality, legal, and political issues [2]. Third, due to the institutional deficiencies in the industry's social responsibility, the relevant laws and regulations are lacking, further indicating that correcting social responsibility deficiencies is weak. Strong legal support for MMRISR management can be enhanced by developing and improving effective policy tools and crafting new laws and rules related to social responsibility. This initiative can help to solve the problem of MIPs' weak sustainable development [27,97].

### 7. Conclusions and Future Research

#### 7.1. Conclusions

Megaproject sustainability encompasses key aspects of social responsibility, environmental protection, and economic profitability. As a major public infrastructure project, MMPI will have far-reaching effects on the politics, economy, society, environment, and public of the region. Therefore, social responsibility, as an important practice for sustainable development of megaprojects, currently plays an essential role in improving environmental conditions and maintaining social stability. Our findings showed that the lack of social responsibility is closely related to the lack of attention to project evaluation and the unclear social responsibility in the predecision stage. This study introduces the concept of social responsibility in MMPI, screens social responsibility indicators for MMRI, and develops an MMRISR evaluation indicator system to provide a new perspective for MMRI governance.

The MMRISR indicator system proposed in this study was established via the methods of literature review, questionnaire survey, and FAPH. First, this study determined the dimensions and primary indicators through a literature review. Second, the MMRISR evaluation indicator system was established by screening 24 social responsibility indicators for MMRI. A questionnaire survey and expert interviews were conducted, covering the four dimensions of economic and quality responsibility, legal responsibility, environmental and ethical responsibility, and political responsibility. Then, FAHP was used to calculate the weights of the social responsibility indicators. Finally, the findings were applied to the Zhengzhou 107 Auxiliary Road Project as the case study to prove the feasibility of the model's application to a practical example.

The research contributions of this study are as follows. First, this study has established the MMRISR evaluation indicator system. This indicator system not only helps to improve the science basis of project establishment and decision-making but also strengthens the attention and implementation of social responsibility of key stakeholders. Moreover, the method and process of establishing the indicator system can provide reference for the establishment of MSR evaluation criteria for other types of projects. Second, this study has developed a weighted indicator system. This approach can overcome the obstacles of quantifying social responsibility issues. The weighting priority can reflect the problems and trends of MMRISR management, and it further provides directions for project decision makers for developing effective systems or policies. Third, with a strong theoretical basis, the indicator system can be subsequently incorporated into regulatory systems (e.g., feasibility studies) and provide a decision-making tool for MMRISR governance. When the social responsibility of implemented projects are evaluated, we can grasp whether the fulfillment of social responsibility deviates from the social responsibility requirements in the prefeasibility study and correct deviations on time to fundamentally promote the sustainable development of MMRI. Finally, this study enriches the theory of social responsibility and sustainability of major projects, and it is conducive to the promotion of sustainable development of MIPs. Similarly, the findings offer a practical value for guiding the government to effectively govern MSR.

### 7.2. Limitations and Future Research

This study has several limitations, but it also reveals promising areas for future research. First, the MMRISR evaluation indicator system was established on the basis of unique Chinese conditions. Differences in cultural background and economic development imply a variety of social responsibility measures for adoption by different countries and regions. Whether the MMRISR evaluation indicator system is applicable to engineering projects in general, or to other countries around the world, remains to be verified by future studies. Second, the complex impact mechanism of social responsibility in MMRI and the numerous stakeholders both require proper attention. In this research, we only studied the social responsibility that should be assumed at the project level. We did not study in depth the interaction and transmission mechanisms among the socially responsible subjects and the social responsibility governance strategies. These topics are worthy of further research.

**Author Contributions:** Conceptualization, D.Y. and J.L.; methodology, J.L. and D.Y.; validation, J.P.; formal analysis, J.L.; investigation, D.Y. and J.P.; resources, L.L. and J.Z.; data curation, J.L. and L.L.; writing—original draft preparation, J.L.; writing—review and editing, J.P. and J.Z.; funding acquisition, D.Y. and J.L. All authors have read and agreed to the published version of the manuscript.

**Funding:** This research was funded by the National Natural Science Foundation of China (Grant-Nos.71801083; 71901220; 71871096; 71901113), Special major projects for research and development of Henan Province (Scientific and technological projects) (GrantNo.212102310002); Graduate Key Project for College of Engineering Management and Real Estate of Henan University of Economics and Law (Grant No.2021-2).

**Conflicts of Interest:** The authors declare no conflict of interest.

## Appendix A. Questionnaire on MMRISR Indicators

| Target Level | Component Level | Level of Endorsement (1 Strongly Disagree; 2 Disagree; 3 Not Necessarily; 4 Agree; 5 Strongly Agree) | | | | |
|---|---|---|---|---|---|---|
| Political responsibility | Compensation for land acquisition and relocation | 1 | 2 | 3 | 4 | 5 |
| | Regional transportation network construction | 1 | 2 | 3 | 4 | 5 |
| | Meets the needs of residents' life and travel | 1 | 2 | 3 | 4 | 5 |
| | Maintaining relationships with the surrounding community | 1 | 2 | 3 | 4 | 5 |
| | Effective coordination between the main government departments | 1 | 2 | 3 | 4 | 5 |
| | Engineering anticorruption | 1 | 2 | 3 | 4 | 5 |
| | Proper handling of the relationship between people, vehicles, roads and the environment | 1 | 2 | 3 | 4 | 5 |
| | Public event emergency response Supplementary | 1 | 2 | 3 | 4 | 5 |
| Financial and Legal Liability | Project economic feasibility decision | 1 | 2 | 3 | 4 | 5 |
| | Project technical feasibility decision | 1 | 2 | 3 | 4 | 5 |
| | Technology innovation and application | 1 | 2 | 3 | 4 | 5 |
| | Sound engineering project governance mechanism | 1 | 2 | 3 | 4 | 5 |
| | Quality and safe construction of road projects | 1 | 2 | 3 | 4 | 5 |
| | Road construction cost and schedule control | 1 | 2 | 3 | 4 | 5 |
| | Road maintenance | 1 | 2 | 3 | 4 | 5 |
| | Project operating costs and safety assurance Supplementary | 1 | 2 | 3 | 4 | 5 |
| Legal Liability | Information disclosure | 1 | 2 | 3 | 4 | 5 |
| | Effective regulation | 1 | 2 | 3 | 4 | 5 |
| | Design, construction, and operation comply with transportation industry specifications and legal requirements | 1 | 2 | 3 | 4 | 5 |
| | Project coverage compliance | 1 | 2 | 3 | 4 | 5 |
| | Project reporting independence and impartiality Supplementary | 1 | 2 | 3 | 4 | 5 |
| Environmental and Ethical Responsibility | Water, noise, dust, and other pollution control | 1 | 2 | 3 | 4 | 5 |
| | Environmental ecological and cultural protection along the road | 1 | 2 | 3 | 4 | 5 |
| | Consideration of the impact of the proposed road on existing tracks Interoperability with the surrounding environment | 1 | 2 | 3 | 4 | 5 |
| | Protecting the rights and interests of participating employees | 1 | 2 | 3 | 4 | 5 |
| | Rational use of construction resources | 1 | 2 | 3 | 4 | 5 |
| | Coordinate the sequence of road construction and pipeline construction | 1 | 2 | 3 | 4 | 5 |
| | Public opinion monitoring | 1 | 2 | 3 | 4 | 5 |
| | Raising awareness of environmental protection | 1 | 2 | 3 | 4 | 5 |
| | Monitoring and reporting wrongdoing Supplementary | 1 | 2 | 3 | 4 | 5 |

## Appendix B. Questionnaire on the Relative Importance of MMRISR Indicators

1. At the level of social responsibility indicators for large municipal road projects, please compare the relative importance of political responsibility $U_1$, economic and quality responsibility $U_2$, legal responsibility $U_3$, environmental and ethical responsibility $U_4$.

| | Extremely Important | Very Important | Obviously Important | Slightly More Important | Equally Important | Slightly Unimportant | Obviously Not Important | Very Unimportant | Extremely Unimportant |
|---|---|---|---|---|---|---|---|---|---|
| $U_1/U_2$ | | | | | | | | | |
| $U_1/U_3$ | | | | | | | | | |
| $U_1/U_4$ | | | | | | | | | |
| $U_2/U_3$ | | | | | | | | | |
| $U_2/U_4$ | | | | | | | | | |
| $U_3/U_4$ | | | | | | | | | |

2. At the level of political responsibility ($U$) for large municipal road projects (1), please compare the relative importance of the following secondary indicators two by two. Construction of regional transportation network, $U_{11}$; meeting residents' living and travel needs, $U_{12}$; maintaining relations with surrounding communities, $U_{13}$; effective coordination among main government departments, $U_{14}$; engineering anticorruption, $U_{15}$; proper handling of relations among people, vehicles, roads and environment, $U_{16}$; emergency response to public events, $U_{17}$.

| | Extremely Important | Very Important | Obviously Important | Slightly More Important | Equally Important | Slightly Unimportant | Obviously Not Important | Very Unimportant | Extremely Unimportant |
|---|---|---|---|---|---|---|---|---|---|
| $U_{11}/U_{12}$ | | | | | | | | | |
| $U_{11}/U_{13}$ | | | | | | | | | |
| $U_{11}/U_{14}$ | | | | | | | | | |
| $U_{11}/U_{15}$ | | | | | | | | | |
| $U_{11}/U_{16}$ | | | | | | | | | |
| $U_{11}/U_{17}$ | | | | | | | | | |
| $U_{12}/U_{13}$ | | | | | | | | | |
| $U_{12}/U_{14}$ | | | | | | | | | |
| $U_{12}/U_{15}$ | | | | | | | | | |
| $U_{12}/U_{16}$ | | | | | | | | | |
| $U_{12}/U_{17}$ | | | | | | | | | |
| $U_{13}/U_{14}$ | | | | | | | | | |
| $U_{13}/U_{15}$ | | | | | | | | | |
| $U_{13}/U_{16}$ | | | | | | | | | |
| $U_{13}/U_{17}$ | | | | | | | | | |
| $U_{14}/U_{15}$ | | | | | | | | | |
| $U_{14}/U_{16}$ | | | | | | | | | |
| $U_{14}/U_{17}$ | | | | | | | | | |
| $U_{15}/U_{16}$ | | | | | | | | | |
| $U_{15}/U_{17}$ | | | | | | | | | |
| $U_{16}/U_{17}$ | | | | | | | | | |

3. At the level of economic and quality responsibility ($U_2$) for large municipal road projects, you are invited to compare the relative importance of the following secondary indicators two by two. Project economic feasibility decision, $U_{21}$; project technical feasibility decision, $U_{22}$; perfect project governance mechanism, $U_{23}$; road project

quality and safety construction, $U_{24}$; road construction cost and schedule control, $U_{25}$; project operation cost and safety guarantee, $U_{26}$.

| | Extremely Important | Very IM-PORTANT | Obviously Important | Slightly More Important | Equally Important | Slightly Unimportant | Obviously Not Important | Very Unimportant | Extremely Unimportant |
|---|---|---|---|---|---|---|---|---|---|
| $U_{21}/U_{22}$ | | | | | | | | | |
| $U_{21}/U_{23}$ | | | | | | | | | |
| $U_{21}/U_{24}$ | | | | | | | | | |
| $U_{21}/U_{25}$ | | | | | | | | | |
| $U_{21}/U_{26}$ | | | | | | | | | |
| $U_{22}/U_{23}$ | | | | | | | | | |
| $U_{22}/U_{24}$ | | | | | | | | | |
| $U_{22}/U_{25}$ | | | | | | | | | |
| $U_{22}/U_{26}$ | | | | | | | | | |
| $U_{23}/U_{24}$ | | | | | | | | | |
| $U_{23}/U_{25}$ | | | | | | | | | |
| $U_{23}/U_{26}$ | | | | | | | | | |
| $U_{24}/U_{25}$ | | | | | | | | | |
| $U_{24}/U_{26}$ | | | | | | | | | |
| $U_{25}/U_{26}$ | | | | | | | | | |

4. At the level of legal responsibility for large municipal road projects ($U_3$), please compare the relative importance of the following secondary indicators two by two. Information disclosure, $U_{31}$; effective regulation, $U_{32}$; design, construction, and operation in accordance with transportation industry norms and legal requirements, $U_{33}$.

| | Extremely Important | Very Important | Obviously Important | Slightly More Important | Equally Important | Slightly Unimportant | Obviously Not Important | Very Unimportant | Extremely Unimportant |
|---|---|---|---|---|---|---|---|---|---|
| $U_{31}/U_{32}$ | | | | | | | | | |
| $U_{31}/U_{33}$ | | | | | | | | | |
| $U_{32}/U_{33}$ | | | | | | | | | |

5. At the level of environmental and ethical responsibility of large municipal road projects ($U_4$), please compare the relative importance of the following secondary indicators two by two. Water, noise, dust, and other pollution control, $U_{41}$; environmental, ecological, and cultural protection along the road, $U_{42}$; consideration of the impact of the proposed road on the existing tracks, $U_{43}$; interoperability with the surrounding environment, $U_{44}$; protection of the rights and interests of the participating employees, $U_{45}$; coordination of the road construction and pipeline construction sequence, $U_{46}$; awareness of environmental protection, $U_{47}$; monitoring and prosecution of illegal acts, $U_{48}$.

| | Extremely Important | Very Important | Obviously Important | Slightly More Important | Equally Important | Slightly Unimportant | Obviously Not Important | Very Unimportant | Extremely Unimportant |
|---|---|---|---|---|---|---|---|---|---|
| $U_{41}/U_{42}$ | | | | | | | | | |
| $U_{41}/U_{43}$ | | | | | | | | | |
| $U_{41}/U_{44}$ | | | | | | | | | |
| $U_{41}/U_{45}$ | | | | | | | | | |

| | Extremely Important | Very Important | Obviously Important | Slightly More Important | Equally Important | Slightly Unimportant | Obviously Not Important | Very Unimportant | Extremely Unimportant |
|---|---|---|---|---|---|---|---|---|---|
| $U_{41}/U_{46}$ | | | | | | | | | |
| $U_{41}/U_{47}$ | | | | | | | | | |
| $U_{41}/U_{48}$ | | | | | | | | | |
| $U_{42}/U_{43}$ | | | | | | | | | |
| $U_{42}/U_{44}$ | | | | | | | | | |
| $U_{42}/U_{45}$ | | | | | | | | | |
| $U_{42}/U_{46}$ | | | | | | | | | |
| $U_{42}/U_{47}$ | | | | | | | | | |
| $U_{42}/U_{48}$ | | | | | | | | | |
| $U_{43}/U_{44}$ | | | | | | | | | |
| $U_{43}/U_{45}$ | | | | | | | | | |
| $U_{43}/U_{46}$ | | | | | | | | | |
| $U_{43}/U_{47}$ | | | | | | | | | |
| $U_{43}/U_{48}$ | | | | | | | | | |
| $U_{44}/U_{45}$ | | | | | | | | | |
| $U_{44}/U_{46}$ | | | | | | | | | |
| $U_{44}/U_{47}$ | | | | | | | | | |
| $U_{44}/U_{48}$ | | | | | | | | | |
| $U_{45}/U_{46}$ | | | | | | | | | |
| $U_{45}/U_{47}$ | | | | | | | | | |
| $U_{45}/U_{48}$ | | | | | | | | | |
| $U_{46}/U_{47}$ | | | | | | | | | |
| $U_{46}/U_{48}$ | | | | | | | | | |
| $U_{47}/U_{48}$ | | | | | | | | | |

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
