# Peer review of "Evaluation of Social Responsibility of Major Municipal Road Infrastructure—Case Study of Zhengzhou 107 Auxiliary Road Project"

_buildings, doi:10.3390/buildings12030369_

Round 1
Reviewer 1 Report
The article "Research on the Evaluation of Social Responsibility of Major Municipal Road Infrastructure" presents an interesting study on indicators for evaluating social responsibility. The research presented in the paper comprehensively describes the approach proposed by the authors. In terms of sustainable development, especially the social aspect addressed in the paper, it is a multidisciplinary study, which is an additional asset of the paper. The methodology is presented accurately and understandably. The authors have also clearly presented the conclusions and insights of the research. I have no major comments on the work.
line 294 there is written 26 indicator, should be 24
Author Response
We appreciate reviewer 1 for his/her effort to review our manuscript, and his/her positive feedback. The reviewer gives an accurate summary of our work and brings forward constructive questions.
We have addressed the error in the number of indicators.
”After the screening, 24 indicators were retained to establish the MMRISR evaluation indicator system.”
Again, we are very grateful to the Reviewer for reviewing the paper so carefully.
Reviewer 2 Report
The publication entitled – "Research on the Evaluation of Social Responsibility of Major 2 Municipal Road Infrastructure", presents scientific research which can be divided into two groups: (i) Author's evaluation system MMRISR, and (ii) road case study Zhengzhou 107, located in Zhengzhou City, Henan Province, China.
In the reviewer's opinion, the publication was prepared correctly. In particular, the methodology of the author's evaluation system should be highlighted. The authors have prepared an extensive literature analysis on the basis of which it was possible to prepare the author's evaluation system. Many evaluation systems researchers end at this stage, in this case it was different. The authors prepared an evaluation of the hierarchy of the obtained indicators, using the fuzzy analytic hierarchy process method.
In the reviewer's opinion, the research should be prepared on a larger study group than 20 people. However, it should be pointed that this would be difficult to implement among MMRISR professionals. However, such a small study group is not authoritative, as can be seen in Table 1, e.g. the study group is mostly male, middle-aged, involved in design.
In the reviewer's opinion, it is worth presenting in an appendix to the publication - the form that the 20 senior managers had to fill in. The reader should be able to see the questions asked and evaluate them for themselves.
In the reviewer's opinion, the publication is acceptable for publication, but a language correction should be prepared.
Author Response
Response to Reviewer 2 Comments
Thank you for your decision and constructive comments on my manuscript. We have carefully considered the suggestion of Reviewer and make some changes. The highlighted red parts in the revised version were modified and added based on your comments. Please see the attachment. I hope this revision can make my paper more acceptable. The revisions were addressed point by point below.
Point 1: In the reviewer's opinion, the research should be prepared on a larger study group than 20 people. However, it should be pointed that this would be difficult to implement among MMRISR professionals. However, such a small study group is not authoritative, as can be seen in Table 1, e.g. the study group is mostly male, middle-aged, involved in design.
Response 1: In fact, a larger sample of surveys would make the results more reliable. As the reviewer stated, this is difficult to implement among MMRISR professionals. Therefore, we made an effort to ensure the authority and reliability of the data.
Due to the limitations of the sample of large construction projects, the questionnaire used in this study was sent to 20 senior managers who had been involved in the planning and establishment of MMRISR to solicit their expert evaluation of social responsibility indicators at the indicator identification and indicator relative importance determination stages. In particular, we ensured that the interviewed experts were involved in the construction of at least one MMRI. To maximize variability and generalizability, we sought out project experts and senior managers of different sizes, located in different geographic locations, and demonstrating different levels of MSR performance. Since the interviewed experts are long-time senior managers in the engineering industry, they have more professional experience in this field to ensure the reliability of the survey, and the gender of the experts is all male which is largely in line with the gender imbalance in the engineering field. The proportion of demographic information of these participant samples basically reflects the actual situation of organizational forms in China.
Point 2: In the reviewer's opinion, it is worth presenting in an appendix to the publication - the form that the 20 senior managers had to fill in. The reader should be able to see the questions asked and evaluate them for themselves.
Response 2: This has been added to the revised version of the manuscript, and a questionnaire completed by 20 senior managers has been included in the revised version. Table A1 Questionnaire on MMRISR indicators and Table A2 Questionnaire on the relative importance of MMRISR indicators have been included in the revision as appendices.
Point 3: In the reviewer's opinion, the publication is acceptable for publication, but a language correction should be prepared.
Response 3: We agree with this suggestion and have modified terminology throughout the text as appropriate. We apologize for the poor language of our manuscript. We worked on the manuscript for a long time. We have now worked on both language and readability and have also involved native English speakers for language corrections. We really hope that the flow and language level have been substantially improved.
In all, I found the reviewer’s comments are quite helpful, and I revised my paper point-by-point. Thank you and the review again for your help!

Reviewer 3 Report
I have read this article. While the topic is most interesting, the paper presents some minor limitations:
The introduction is not very useful. Therefore, the introduction should be extended very carefully. The introduction section should be rewritten again. The introduction should highlight the study's novelty and motivation and put some literature without any useful explanation.
I would suggest the author improve their theoretical discussion and arrives at their debate or argument. In addition, the background introduction should be condensed. The literature review is not presented in a good structure.
There are several grammatical errors in the paper. Proof-read suggested.
Author Response
Response to Reviewer 3 Comments
Thank you for your decision and constructive comments on my manuscript. We have carefully considered the suggestion of the Reviewer and make some changes. The red parts in the revised version were modified and added based on your comments. Please see the attachment. I hope this revision can make my paper more acceptable. The revisions were addressed point by point below.
Point 1: The introduction is not very useful. Therefore, the introduction should be extended very carefully. The introduction section should be rewritten again. The introduction should highlight the study's novelty and motivation and put some literature without any useful explanation. I would suggest the author improve their theoretical discussion and arrives at their debate or argument. In addition, the background introduction should be condensed. The literature review is not presented in a good structure.
Response 1: In response to the reviewers' comments, we have revised the introduction, theoretical discussion, background, and literature review. First, the introduction was carefully extended. The novelty and motivation were highlighted, the background was condensed as well as the structure of the literature review was revised and content was added. Then, we have improved for the theoretical discussion to arrive at our debate.
Point 2: There are several grammatical errors in the paper. Proof-read suggested.
Response 2: We agree with this suggestion and have modified terminology throughout the manuscript as appropriate. We apologize for the grammatical errors in our manuscript. We worked on the manuscript for a long time. We have now worked on both language and readability and have also involved native English speakers for language corrections. We really hope that the flow and language level have been substantially improved.
In all, I found the reviewer’s comments are quite helpful, and I revised my paper point-by-point. Thank you and the review again for your help!

Reviewer 4 Report
The article provides an interesting methodology for assessing the impact of social responsibility for a sustainable management of infrastructures.
The results are clearly presented according to a reasonable scientific design.
All areas of concern have been addressed according to standardized procedures.
The statistical significance of the analysis is limited to a very focused population selection. Comments on alternatives and possible impact of this limited approach are required.
On the conclusion an adequate foundation should be presented for explaining why the conceptual framework is adequate specifically for urban megaprojects and their sustainability should be added.
Author Response
Response to Reviewer 4 Comments
Thank you for your decision and constructive comments on my manuscript. We have carefully considered the suggestion of the Reviewer and make some changes. We have tried our best to improve and made some changes to the manuscript. The highlighted parts in the revised version were modified and added based on your comments. Please see the attachment. I hope this revision can make my paper more acceptable. The revisions were addressed point by point below.
Point 1: The statistical significance of the analysis is limited to a very focused population selection. Comments on alternatives and possible impact of this limited approach are required.
Response 1: In order to take this reviewer concern into account, and improve the quality of our manuscript, the possible impact of limiting the statistical significance of the analysis to a very concentrated population is discussed in the revised version of the manuscript, and this is further supplemented.
In fact, a larger sample of surveys would make the results more reliable. As the reviewer stated, this is difficult to implement among MMRISR professionals. Therefore, we made an effort to ensure the authority and reliability of the data.
Due to the limitations of the sample of megaprojects, the questionnaires used in this study were sent to 20 senior managers who had participated in the planning and establishment of MMRI to solicit an expert evaluation of social responsibility indicators at the stage of indicator determination and indicator relative importance judgment. Therefore, the established system of social responsibility evaluation indicators is only for the MMRI field.
Meanwhile, social responsibility indicators are highly subjective. Thus, the data sources in this study were strictly controlled during indicator selection. To maximize variability and generalizability, we sought out megaproject experts and senior managers of different sizes, located in different geographic locations, and demonstrating different levels of MSR performance. In particular, we ensured that the interviewed experts had participated in the construction of at least one MMRI. The experts are all highly educated, most of them have a master’s degree or a higher educational attainment, and have intermediate or higher titles. The experts also include the following: those with government roles; owners; part of design, construction, supervision, and operation units; and those who had participated in the project establishment, design, construction and operation, and maintenance stages. Since the interviewed experts are senior managers with long experience in the engineering industry, they have more professional experience in this field to ensure the reliability of the survey, and the gender of the experts is all male, which basically conforms to the characteristics of gender imbalance in the engineering field. These data proportions basically reflect the actual situation of organizational forms in China. The data indicate that the interviewed experts have experience and knowledge of the issues related to this study, which adds to the credibility of the data. Furthermore, ten experts who had participated in the construction of MMRI were invited to judge the MMRISR evaluation indicator system in terms of the degree of recognition of the indicators.
Point 2: On the conclusion an adequate foundation should be presented for explaining why the conceptual framework is adequate specifically for urban megaprojects and their sustainability should be added.
Response 2: This has been complemented in a revised version of the manuscript, and the explanation of conceptual framework is particularly suitable for major municipal road infrastructure(MMRI) and their sustainability has been added to the revised version.
This study introduces the concept of social responsibility in MMPI, screens social responsibility indicators for MMRI and develops an MMRISR evaluation indicator system to provide a new perspective for MMRI governance. Therefore, the social responsibility evaluation indicator system constructed for the characteristics of MMRI is applicable to major municipal infrastructure projects.
Megaproject sustainability encompasses key aspects of social responsibility, environmental protection, and economic profitability. As a major public infrastructure project, MMPI will have far-reaching effects on the politics, economy, society, environment, and public of the region. Therefore, social responsibility, as an important practice for sustainable development of megaprojects, currently plays an essential role in improving environmental conditions and maintaining social stability. Thus, the fulfillment of megaproject social responsibility can significantly improve the sustainability of major municipal infrastructure.
In all, I found the reviewer’s comments are quite helpful, and I revised my paper point-by-point. Thank you and the review again for your help!

Round 2
Reviewer 4 Report
All the points raised on the initial review have been addressed and can be accepted for publishing.
Author Response
Thank you for your response and comments and approval of our paper entitled "Research on the Evaluation of Social Responsibility of Major Municipal Road Infrastructure". Those comments are all valuable and very helpful for revising and improving our paper, as well as the important guiding significance to our research.
In all, thanks again to the reviewer for your help!
This manuscript is a resubmission of an earlier submission. The following is a list of the peer review reports and author responses from that submission.